# Circulating Nucleosomes as a Novel Biomarker for Sepsis: A Scoping Review

**DOI:** 10.3390/biomedicines12071385

**Published:** 2024-06-21

**Authors:** Fuhong Su, Anthony Moreau, Marzia Savi, Michele Salvagno, Filippo Annoni, Lina Zhao, Keliang Xie, Jean-Louis Vincent, Fabio Silvio Taccone

**Affiliations:** 1Laboratoire de Recherche Experimentale des Soins Intensifs, Hôpital Universitaire de Bruxelles (HUB), Université Libre de Bruxelles, 1070 Brussels, Belgium; anthony.moreau@ulb.be (A.M.); filippo.annoni@hubruxelles.be (F.A.); jlvincent@intensive.org (J.-L.V.); fabio.taccone@hubruxelles.be (F.S.T.); 2Department of Intensive Care, Hôpital Universitaire de Bruxelles (HUB), Université Libre de Bruxelles, 1070 Brussels, Belgium; marzia.savi@humanitas.it (M.S.); michele.salvagno@ulb.be (M.S.); 3Department of Anesthesiology and Intensive Care, IRCCS Humanitas Research Hospital, 20089 Milan, Italy; 4Department of Critical Care Medicine, Tianjin Medical University General Hospital, Tianjin 300052, China; 18240198229@163.com (L.Z.); xiekeliang2009@hotmail.com (K.X.)

**Keywords:** nucleosome, NETosis, sepsis, infection, sepsis diagnosis, prognosis

## Abstract

Circulating nucleosome levels are commonly elevated in physiological and pathological conditions. Their potential as biomarkers for diagnosing and prognosticating sepsis remains uncertain due, in part, to technical limitations in existing detection methods. This scoping review explores the possible role of nucleosome concentrations in the diagnosis, prognosis, and therapeutic management of sepsis. A comprehensive literature search of the Cochrane and Medline libraries from 1996 to 1 February 2024 identified 110 potentially eligible studies, of which 19 met the inclusion criteria, encompassing a total of 39 SIRS patients, 893 sepsis patients, 280 septic shock patients, 117 other ICU control patients, and 345 healthy volunteers. The enzyme-linked immunosorbent assay [ELISA] was the primary method of nucleosome measurement. Studies consistently reported significant correlations between nucleosome levels and other NET biomarkers. Nucleosome levels were higher in patients with sepsis than in healthy volunteers and associated with disease severity, as indicated by SOFA and APACHE II scores. Non-survivors had higher nucleosome levels than survivors. Circulating nucleosome levels, therefore, show promise as early markers of NETosis in sepsis, with moderate diagnostic accuracy and strong correlations with disease severity and prognosis. However, the available evidence is drawn mainly from single-center, observational studies with small sample sizes and varied detection methods, warranting further investigation.

## 1. Introduction

Sepsis remains a significant global health concern, responsible for approximately 20% of deaths worldwide, with septic shock mortality rates reaching close to 60% [1]. Biomarkers can be used to indicate the presence of sepsis, its severity, and its response to treatment [2,3]. Despite the identification of more than 250 potential biomarkers of sepsis [4], only two, the host-response markers C-reactive protein (CRP) and procalcitonin (PCT), are widely used in clinical practice and these are not specific for sepsis as levels can be raised in other conditions in critically ill patients; as such, serial values are more useful than single measurements. The complexity of sepsis makes it unlikely that a single biomarker will be relevant for all patients at all times and further evaluation and validation is needed to determine the clinical utility of individual biomarkers for specific purposes, including diagnosis, prognosis, and therapeutic guidance. Here, we will discuss the potential role of circulating nucleosomes as biomarkers of sepsis.

Chromatin is a substance composed of DNA and proteins that form chromosomes within the nucleus of cells. The fundamental structural unit of chromatin in all eukaryotic cells is the nucleosome, comprising the following two main components: histones and DNA, as depicted in Figure 1. The histone portion offers structural support and consists of two copies each of the histone proteins H2A, H2B, H3, and H4, encircling approximately 145–147 base pairs of DNA for about 1.65 times. These nucleosomes are interconnected by “linker DNA” segments, approximately 20–80 base pairs long, and bound by histone H1 [5]. This intricate structure regulates chromatin compaction/condensation and governs transcriptional access to the nucleosome [6]. Biochemical and structural studies have elucidated the dynamic nature of nucleosomes, which are essential for gene transcription regulation, DNA replication, repair processes, and efficient higher-order chromatin compaction [5,7,8]. Various molecules and mechanisms, including DNA breathing [7], post-translational histone modifications, histone chaperones, histone variants, and chromatin remodelers, contribute to this dynamic regulation [9,10].

Nucleosomes are released during cellular damage and cell death [11], and circulating levels are increased in physiological conditions, including aging [12], physical exercise [13], and stress [14], as well as pathological conditions, such as aging-related degenerative disorders [15], inflammatory responses, autoimmune diseases, ischemic stroke, trauma, and malignancies [12]. These nucleosomes have a short half-life and are typically removed from circulation within 10 min, primarily degraded by endonucleases in the blood, metabolized in the liver, or eliminated by macrophages and immune cells [16,17,18]. Notably, circulating nucleosomes, along with post-translational histone modifications and specific tumor markers, can help in the diagnosis of certain cancers [19], and monitoring their changes during cancer treatment can be useful in assessing therapeutic efficacy [19].

Studies in animal models of sepsis [18] and trials with healthy volunteers receiving lipopolysaccharides [LPSs] [20] have observed increased nucleosome concentrations, indicative of nucleosome release in response to innate immune cell activation. In sepsis, neutrophil activation triggers the release of granule proteins and chromatin, forming neutrophil extracellular traps [NETs] through a process termed NETosis [21]. This process, along with other forms of increased cell death, such as apoptosis [22], necrosis [23], and pyroptosis [24,25], contribute to nucleosome release into the extracellular space. Core histone proteins, including H3, H4, H2A, and H2B, have been identified as major components of NETs, emphasizing their significance in septic pathophysiology [26].

Circulating nucleosome levels may increase in sepsis due to several factors as follows: (a) increased release of circulating free DNA [cfDNA] leading to a biphasic nucleosome release. The first phase is marked by the contribution of cell death within hematopoietic cells, and the second phase by release from non-hematopoietic cells, such as epithelial and endothelial cells [11,27,28]. Indeed, immune [e.g., neutrophils, monocytes, macrophages, mast cells, dendritic cells, eosinophils, basophils] and parenchymal [e.g., endothelial cells] cell death is reported in sepsis [29,30,31]; (b) insufficient clearance due to suppressed or decreased deoxyribonuclease activity [32,33] and the diminished phagocytosis of apoptotic cells [34]; (c) the release of histones, which induces direct cellular toxicity, prompting immune responses, inflammation, and further cellular injury and death, leading to the amplification of nucleosome cascades [35]; and (d) the binding of acute phase proteins, such as CRP, to positively charged histone components, impeding the removal of circulating nucleosomes [36].

## 2. Immunostimulatory Role of Nucleosomes

Circulating nucleosomes are potent triggers of immune responses [37]. Both components of the nucleosome–double-stranded DNA [dsDNA] and histones–exhibit diverse immunostimulatory effects, both in vitro and in vivo. Histones induce cytotoxicity and proinflammatory signaling through Toll-like receptors [TLRs] 2 and 4, whereas DNA triggers signaling through TLR9 and intracellular nucleic acid sensing mechanisms [38]. Histones are cytotoxic to endothelial cells and promote coagulation by activating platelets, impairing anticoagulant pathways, and inhibiting fibrinolysis [39,40]. The administration of purified histones in mice mirrors human sepsis, with thrombocytopenia, neutrophil migration, and organ failure developing [41]. DNA, which is highly immunogenic, represents a crucial pathogen-associated molecular pattern [PAMP] during infection [42]. It can initiate coagulation via the intrinsic pathway and inhibit fibrinolysis [43,44]. However, the in vivo administration of DNA to healthy or septic mice showed no harmful effects [45].

In some clinical studies, the terms histones and nucleosomes are used interchangeably because of detection method limitations [38]. However, in this review, we focus specifically on studies that measure nucleosome concentrations in sepsis. Following their release into the extracellular space, nucleosomes are internalized by various mammalian cells through multiple endocytic pathways [46]. Cellular uptake is facilitated by electrostatic interactions between histone N-terminal tails and cell surface ligands, followed by clathrin- and caveolae-dependent endocytosis [46]. Nucleosomes induce the direct activation of human neutrophils, triggering CD11b/CD66b upregulation, interleukin [IL]-8 secretion, and increased phagocytic activity, independent of the TLR2/TLR4 pathway [47,48,49]. Studies on the immunostimulatory role of nucleosomes suggest cell-type dependence. Purified nucleosomes at physiological concentrations activate human neutrophils [37], induce endothelial [38] and lymphocyte cell death [50], accelerate microglial inflammation [51], and activate dendritic cells and neutrophils to secrete cytokines [52,53]. Additionally, nucleosome high-mobility group box 1 protein [HMGB1] complexes stimulate immune responses via TLR4 and the receptor for advanced glycation end-products [RAGEs], inducing the secretion of proinflammatory cytokines and expression of costimulatory molecules [54,55]. These findings underscore the diverse and intricate immunostimulatory effects of circulating nucleosomes in various cellular contexts. The involvement of nucleosomes in the processes of inflammation and sepsis is illustrated in Figure 2.

## 3. Nucleosome Administration in Sepsis

Despite their immunostimulatory role, the administration of nucleosomes in sepsis does not appear to be toxic [16]. No cytotoxicity was observed, even at high doses when nucleosomes were injected into healthy or septic mice [15,45]. Intact nucleosomes themselves are not procoagulant, unlike the individual purified components of DNA and histones, which exhibit procoagulant properties [14].

Given the lack of direct toxicity associated with nucleosomes, targeting nucleosomes themselves may not be a valid therapeutic focus in sepsis. Instead, therapeutic strategies typically target histones or related signaling pathways [56]. These strategies include inhibiting histone release or NETosis [57], neutralizing histones with anti-histone monoclonal antibodies [58], or blocking signaling pathways, such as TLRs [59]. Targeting histones could potentially decrease circulating nucleosome levels by mitigating the amplification cascade effect and leading to decreased cell death. A translational study in an ewe septic shock model demonstrated that targeting histones using the histone-neutralizing polyanion sodium-β-O-methyl cellobioside sulfate [mCBS] resulted in decreased circulating nucleosome levels [60].

Elevated circulating nucleosome levels have been observed to correlate with sepsis severity and outcome. In liver transplant patients, nucleosome levels were associated with the occurrence of acute kidney injury, early allograft dysfunction, and early mortality after transplantation [61]. Similarly, raised nucleosome levels after graft reperfusion were associated with the occurrence of systemic inflammatory response syndrome [61].

## 4. Unanswered Questions

Despite the known increase in circulating nucleosome levels in sepsis, several questions remain unanswered as follows:Can circulating nucleosome concentrations serve as a biomarker for NET formation in sepsis?Are circulating nucleosome concentrations diagnostic biomarkers for sepsis?Can circulating nucleosome concentrations serve as markers of organ dysfunction or disease severity in sepsis?Are circulating nucleosome concentrations prognostic biomarkers in sepsis?Can nucleosome levels be used to guide sepsis therapy?

We conducted a scoping review to address these unanswered questions and provide insights into the utility of circulating nucleosomes in the context of sepsis diagnosis, prognosis, and therapy.

## 5. Methods

The Preferred Reporting Items for Systematic Reviews and Meta-Analyses [PRISMA 5.15] guidelines [62] were employed for this review, with an extension for scoping reviews [PRISMA-ScR] [63]. A systematic search of the Cochrane and Medline libraries from 1996 to 1 February 2024 was performed to identify studies that investigated the role of circulating nucleosomes in NETosis in differentiating patients with sepsis from healthy volunteers, those with systemic inflammatory response syndrome [SIRS] without infection, or other ICU patients, or assessed associations between circulating nucleosome levels and disease severity or prognosis. The search used the following keywords: nucleosome AND/OR infection AND/OR sepsis AND/OR septic shock AND/OR NETosis. Studies involving animals, patients without probable infection, and healthy volunteers receiving lipopolysaccharide [LPS] were excluded. Studies in children younger than 28 days were also excluded due to variations in etiology and prognosis between early- and late-onset sepsis in neonates compared to adults [64]. Additionally, studies in adults and children with acute respiratory distress syndrome [ARDS] were excluded because ARDS can be sepsis- or non-sepsis-related [65].

Two independent investigators [FS, AM] extracted patient and study characteristics, and any discrepancies were resolved through consensus.

## 6. Results

Our initial search yielded 110 studies, of which 19 met the inclusion criteria, including data from a total of 39 patients with SIRS, 893 patients with sepsis, 280 patients with septic shock, another 117 ICU patients, and 345 healthy volunteers [18,66,67,68,69,70,71,72,73,74,75,76,77,78,79,80,81,82,83]. The study selection flowchart is shown in Figure 3. Eleven studies were prospective, and two were multicenter (Table 1).

The enzyme-linked immunosorbent assay (ELISA) was the primary method for measuring plasma/serum nucleosome levels; commercial and homemade kits were used with the Cell Death Detection ELISA kit (Roche, Mannheim, Germany) most commonly employed. This kit does not offer a standard curve for objective nucleosome quantification, so the results are expressed in arbitrary units [AU].

Among the nineteen studies, eight investigated the correlation between admission nucleosome levels and markers of NET formation [18,69,73,74,75,78,79,80]. Seven of the eight studies reported a positive association [69,73,74,75,78,79,80] (Table 2). Various NET markers were studied, including citrullinated H3R8-nucleosomes [69], citrullinated histones [69], human neutrophil elastase DNA [74], elastase–α1-antitrypsin complexes [18,78,80], neutrophil elastase [NE] [69,79] and myeloperoxidase [MPO] [69,75].

Thirteen studies explored nucleosomes as a potential diagnostic marker in sepsis [66,67,70,71,72,73,74,77,78,79,81,82,83]; all except one [72] reported significant differences in circulating nucleosome levels on admission between septic patients and healthy volunteers, patients with SIRS, or other ICU patients. Three studies reported areas under the receiver operating characteristic curves [AUCs] for diagnosing sepsis and comparing sepsis patients with non-septic control patients, with values ranging from 0.63 to 0.88 [70,73,81]. One study reported a specificity of 100% and sensitivity of 67–78% [70], and a second study reported a sensitivity of 64% and specificity of 76% with the best nucleosome cut-off of 2.09 AU [81].

Among the nine studies that measured nucleosome levels over time [66,68,69,73,74,77,80,81,83], six reported correlations with disease severity, as measured using the sequential organ failure assessment [SOFA] and/or Acute Physiology, Age and Chronic Health Evaluation [APACHE II] scores [Table 3] [66,69,73,80,81,83]; three studies reported no association [68,74,77].

Thirteen studies reported early differences in circulating nucleosome concentrations in survivors and non-survivors [66,67,68,69,73,74,76,77,78,79,80,81,83], eight of which showed significant differences in admission nucleosome levels between the groups [66,68,73,74,76,78,79,80] (Table 4). Three studies reported AUCs for predicting mortality, ranging from 0.63 to 0.75 [66,68,73] (Table 4).

No study specifically investigated the use of circulating nucleosome levels to guide sepsis therapy.

## 7. Discussion

The main findings, in relation to the study questions listed earlier, can be summarized as follows:Can circulating nucleosome concentrations serve as a biomarker for NET formation in sepsis? Circulating nucleosome concentrations are reliable biomarkers for NETosis in sepsis, reflecting the release of chromatin into the extracellular space as part of the immune response.Are circulating nucleosome concentrations diagnostic biomarkers for sepsis?Nucleosome concentrations have moderate utility as diagnostic biomarkers for sepsis, with limited sensitivity and specificity.Can circulating nucleosome concentrations serve as markers of organ dysfunction or disease severity in sepsis?There is a correlation between circulating nucleosome levels and the severity of sepsis, making them a potential biomarker for the stratification of disease severity.Are circulating nucleosome concentrations prognostic biomarkers in sepsis?Nucleosome concentrations on admission serve as good prognostic biomarkers for predicting mortality within 28 to 30 days.Can nucleosome levels be used to guide sepsis therapy?Currently, there is no evidence to support the use of circulating nucleosomes to guide therapy in sepsis.

The activation of neutrophils and the subsequent release of chromatin into the extracellular space as NETs plays a crucial role in immune response during sepsis. NETosis is triggered by pathogen-associated molecular patterns [PAMPs] from microbes or damage-associated molecular patterns [DAMPs] from damaged tissues. NETs can help trap and kill bacteria and can also contribute to tissue damage and organ dysfunction by activating blood coagulation, impairing fibrinolysis, and injuring endothelial cells. Markers of NET release include various molecules, such as cell-free DNA [cf-DNA], nucleosomes, histones, neutrophil elastase, myeloperoxidase, calprotectin, cathelicidins, defensins, and actin. Among these markers, circulating nucleosome levels have been of particular interest due to their association with NETosis. In healthy individuals, an intravenous injection of *Escherichia coli* LPS resulted in a sharp increase in circulating nucleosome levels, peaking approximately 3 h post-injection before rapidly decreasing [18,20]. In septic patients, one small retrospective study showed that nucleosome concentrations were similar in those with [>100 ng/mL, n = 8] and without [<100 ng/mL, n = 12] neutrophil activation on admission, with no correlation between the nucleosome and elastase–a1-antitrypsin complex levels [18]. However, in the same study, nucleosome levels correlated positively with elastase–a1-antitrypsin complex levels in LPS-challenged healthy volunteers [18], suggesting that nucleosome levels’ ability to detect NETosis may depend on the duration and severity of the sepsis. Nevertheless, all the other seven studies [69,73,74,75,78,79,80] that compared the admission of nucleosome concentrations and markers of NET formation in patients with sepsis reported a positive correlation. Larger-scale studies are needed to provide further insight into the relationship between circulating nucleosome levels and NETosis markers in patients with sepsis.

The search for reliable biomarkers for sepsis has been challenging among the studies included in our analysis, with 13 compared admission circulating nucleosome levels in healthy volunteers or control ICU patients and patients with sepsis. All except one biomarker showed a significant difference in nucleosome concentrations in the two groups of subjects, with three studies reporting AUCs for sepsis diagnosis between 0.63 and 0.88 [70,73,81], indicating moderate diagnostic accuracy.

Despite the limited specificity and sensitivity of nucleosome concentrations as a diagnostic biomarker for sepsis, they may still serve as a useful stratification tool in sepsis. The studies that monitored nucleosome levels over time found a strong correlation between nucleosome concentrations and sepsis severity [66,69,80,81]. In an observational study of 50 healthy volunteers and 151 patients with sepsis, the presence of combined high nucleosome and IL-6 values at admission identified a subset of patients who died rapidly [66], suggesting that high nucleosome levels may indicate a hyper-inflammatory response; such patients may need more aggressive anti-inflammatory treatments. In summary, although nucleosome concentrations may not be ideal as a standalone diagnostic biomarker for sepsis because of their limited specificity and sensitivity, they hold promise as stratification markers and, thus, can possibly help guide sepsis treatment. Further research with larger studies is needed to explore the potential of circulating nucleosome levels for guiding treatment strategies in septic patients.

The potential benefit of targeting circulating nucleosomes in sepsis therapeutics remains uncertain, as evidenced by conflicting findings from in vivo studies [45,60]. The administration of histones to mice with sepsis led to increased levels of markers of inflammation and coagulation, suggesting a potentially detrimental effect [41]. However, similar effects were not observed in sham or septic mice who were administered DNA or nucleosomes, suggesting that the harmful effects may be specific to histones [45]. Moreover, a recent translational study conducted in an ovine septic shock model reported promising results regarding the neutralization of circulating histones [60]; in this study, the neutralization of circulating histones appeared to interrupt the harmful amplification cycle induced by increased histone levels. This intervention resulted in decreased inflammation, reduced vasopressor requirements, improved tissue perfusion, mitigated multi-organ dysfunction, and ultimately increased survival rates. These findings suggest that while circulating nucleosomes, particularly histones, may contribute to the pathogenesis of sepsis, targeted interventions aimed at neutralizing or reducing their levels could potentially offer therapeutic benefits.

The variation in methods and kits for measuring nucleosome levels across centers and studies poses challenges in reaching a consensus regarding thresholds for quantifying the presence of NETs in sepsis and septic shock. Some tests have been unable to differentiate between SIRS and sepsis, highlighting the need for standardized and reliable assays. The Cell Death Detection ELISAPLUS kit [Roche] was widely used in the studies in this review, although it was initially designed for apoptosis assessment [84] and only later modified for nucleosome evaluation [85]. Additionally, serum nucleosome levels have been reported to be higher than plasma levels with ELISA tests [86], potentially due to the clotting process, although there were no direct comparisons in the included studies. Newer commercial kits using a chemiluminescence immunoassay performed on an automated immunoanalyzer offer several advantages. They provide rapid and reliable quantification, enabling circulating nucleosomes and their post-translational modifications to be monitored in real-time or near-real-time. Standardized and automated methods can facilitate large-scale routine testing in clinical practice, potentially leading to more accurate, reliable, and homogeneous data across different patient populations. We showed, using an ovine septic shock model [60], that the Nu.Q H3.1 ELISA sandwich assay [Volition, Isnes, Belgium] performed better than the Cell Death Detection ELISAPLUS kit [Roche], providing more reliable measurements: the co-efficiency correlation between the two kits was 0.33 [95% CI: 0.19 to 0.47 *p* < 0.0001].

Our study has several limitations that should be considered when interpreting the results. First, all the studies included in our analysis were observational with limited patient cohort sizes, and most were conducted at single centers, leading inherently to potential bias and limiting the generalizability of the findings to broader populations. Second, most studies had small sample sizes, which may limit the statistical power and generalizability of the results. Larger sample sizes are necessary to obtain more robust and reliable findings. Third, different methods were used to measure nucleosome levels across the studies, making it challenging to make comparisons and impossible to conduct formal meta-analyses. This heterogeneity complicates the interpretation of our results. To address these limitations and provide more conclusive evidence, future research should aim for prospective, large-scale, multicenter clinical studies with standardized measurement methods. Such studies would offer greater statistical power, increased generalizability, and more homogenous findings, ultimately advancing our understanding of the role of circulating nucleosomes in sepsis and their potential clinical implications.

## 8. Conclusions

Circulating nucleosome levels show promise as a biomarker for NETosis in sepsis, particularly in the early stages of the condition. While their diagnostic performance for sepsis is only moderate, they exhibit stronger correlations with sepsis severity and prognosis. Moreover, circulating nucleosome levels hold potential as a target for personalized treatment in sepsis management. However, further evaluation through large-scale randomized clinical trials is necessary to validate these findings and definitively determine their clinical utility.

## Figures and Tables

**Figure 1 biomedicines-12-01385-f001:**
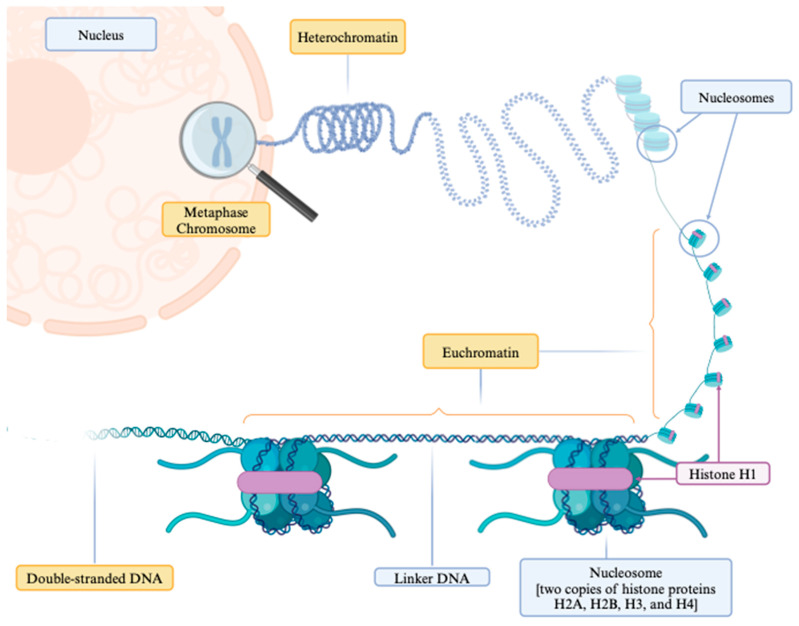
The structure of chromatin within a cell nucleus. This illustration depicts chromatin unfolding to expose euchromatin regions, characterized by a relaxed structure conducive to transcriptional processes. Nucleosomes are complexes of DNA coiled about 1.65 times around core histone proteins: two copies each of the histone proteins H2A, H2B, H3, and H4, bound together and secured by the histone H1. These nucleosomes are interconnected by segments of DNA known as “linker DNA”. The images were created using https://www.biorender.com (accessed on 10 April 2024).

**Figure 2 biomedicines-12-01385-f002:**
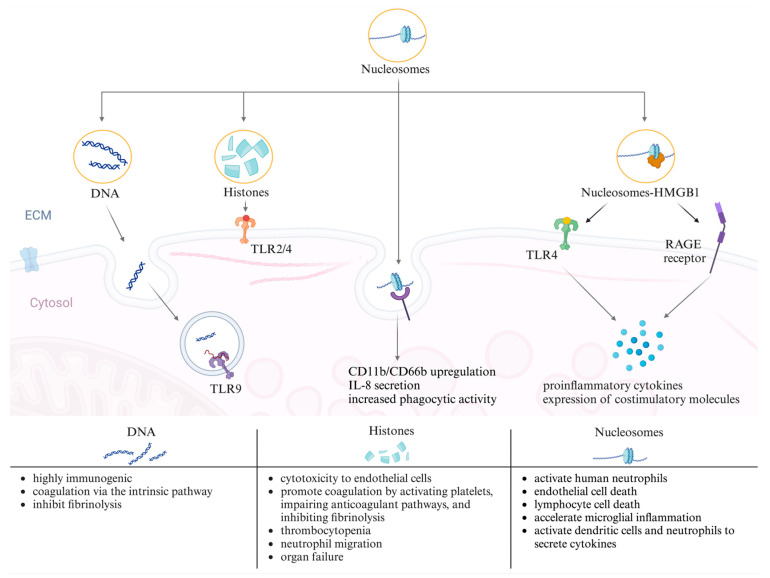
Visualization of the role of nucleosomes in the pathogenesis of inflammation and sepsis DNA, histones, and the nucleosome complex with the high-mobility group box 1 protein [HMGB1] and engaging with cell surface receptors like Toll-like receptor [TLR]2/4 and the receptor for advanced glycation end-products [RAGEs], as well as intracellular receptors such as TLR9, initiating signaling cascades. The consequences of these interactions include the up-regulation of CD11b/CD66b, secretion of interleukin [IL]-8, increased phagocytic activity, and the release of proinflammatory cytokines. These events contribute to several effects, as described at the bottom of the figure. The images were created using https://www.biorender.com (accessed on 10 April 2024).

**Figure 3 biomedicines-12-01385-f003:**
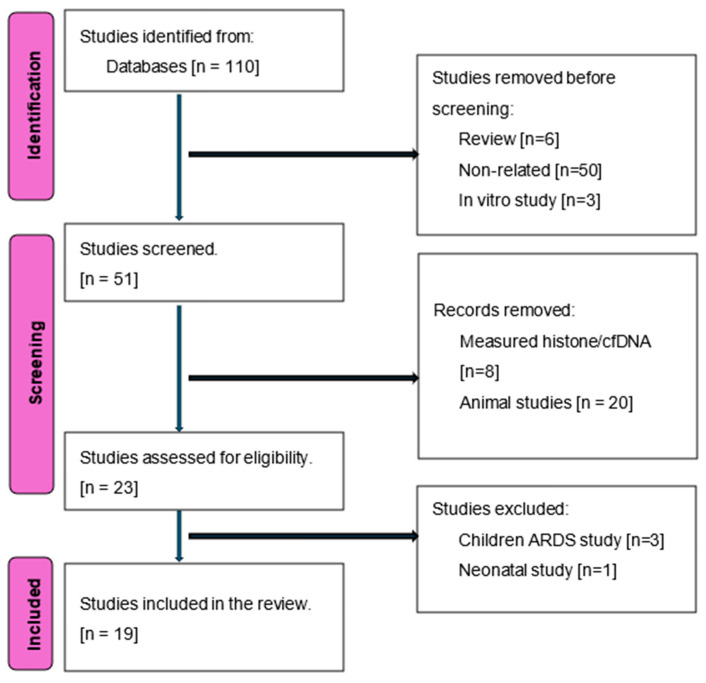
Flowchart of included studies.

**Table 1 biomedicines-12-01385-t001:** Studies included in the current review.

Author [Ref.]	Year	Study Type	Sample	Methods	Catching Ab Detection Ab	Patient Population	Range
Haem Rahimi et al. [66]	2023	Retrospective Monocenter	Plasma	Chemiluminescence immunoassay [Volition]	Nucleosomes H3.1 [H2A, H3B, H3, H4 + DNA]	50 healthy volunteers 151 septic shock	Median 15.4 ng/mL Day 1–2 median 1515 ng/mL
Eichhorn et al. [67]	2023	Prospective Monocenter	Plasma	ELISA ROCHE	Mono anti-histone Mono anti-DNA-POD	25 healthy volunteers 78 sepsis patients [14 with COVID]	0.01 [0.01; 0.02] AU 0.09 [0.05; 0.11] AU without COVID 0.11 [0.04; 0.15] AU with COVID
Rai et al. [68]	2022	Prospective Monocenter	Serum	ELISA [Orgentec]	Unknown	80 sepsis	male: 209.8 [68.0–1263.0] ρg/μL female: 248.7 [65.0–1721.0] ρg/μL
Morimont et al. [69]	2022	Retrospective Monocenter	Plasma	ELISA [Volition]	Nucleosomes H3.1 [H2A, H3B, H3, H4 + DNA]	48 control patients 46 septic shock 22 critical COVID-19	24.6 [12.2–61.7] ng/mL 862 [252–9398] ng/mL
Beltrán-García et al. [70]	2021	Retrospective Monocenter	Plasma	ELISA Kit 1 [home made] Kit2 [Roche]	Mono anti-histone Mono anti-DNA-POD	17 healthy volunteers 9 ICU control patients 10 septic ICU patients 17 septic shock	70.66 ± 42.22 ng/mL (kit 1); 0.083 ± 0.04 AU (kit 2) 56.97 ± 25.76 ng/mL (kit 1); 0.080 ± 0.01 AU (kit 2) 111.8 ± 74.50 ng/mL (kit 1); 0.130 ± 0.08 AU (kit 2) 152.7 ± 74.93 ng/mL (kit 1); 0.216 ± 0.17 AU (kit 2)
van der Meer et al. [18]	2019	Retrospective Monocenter	Plasma	ELISA	Mono anti-histone mAb CLB-ANA/58	20 sepsis patients	NR [only figure available]
Patel et al. [71]	2019	Retrospective Monocenter	Plasma	ELISA [Roche]	Mono anti-histone Mono anti-DNA-POD	50 healthy volunteers 20 sepsis + no DIC 59 sepsis + non-overt DIC 24 sepsis + overt DIC	<10AU <10 AU 10–15 AU 20–30 AU
Lee et al. [72]	2018	Prospective Monocenter	Plasma	ELISA [Roche]	Mono anti-histone Mono anti-DNA-POD	21 sepsis patients 23 healthy volunteers	0.3 ± 0.08 U/L 0.1 ± 0.03 U/L
Duplessis et al. [73]	2018	Retrospective Multicenter [4 in USA]	Plasma	ELISA [Roche]	Mono anti-histone Mono anti-DNA-POD	24 non-infectious SIRS 4 uncomplicated sepsis 127 severe sepsis 35 septic shock	1.1 ± 1.7 µg/mL 1.7 ± 1.9 µg/mL 3.0 ± 9.4 µg/mL 5.5 ± 10.9 µg/mL
Kaufman et al. [74]	2017	Prospective Monocenter	Plasma	ELISA [Roche]	Mono anti-histone Mono anti-DNA-POD	30 healthy volunteers 24 sepsis	0 [0–0.1] µg/mL 0.35 [0–1.9] µg/mL
Delabranche et al. [75]	2017	Prospective Monocenter	Plasma	ELISA [Roche]	Mono anti-histone Mono anti-DNA-POD	20 septic shock [10 with DIC vs. 10 without]	Higher in patients with DIC
Raffray et al. [76]	2015	Prospective Monocenter	Plasma	ELISA [Roche]	Mono anti-histone Mono anti-DNA-POD	17 healthy volunteers 49 septic shock 22 cardiogenic shock	NR [only figure available]
Miki et al. [77]	2015	Prospective Monocenter	Plasma	ELISA [Roche]	Mono anti-histone Mono anti-DNA-POD	5 healthy volunteers 30 sepsis patients [20 survivors, 10 non-survivors]	NR [only figure available]
Huson et al. [78]	2015	Prospective Monocenter	Plasma	ELISA [Sanquin]	Monoclonal antibody H3 Monoclonal antibody nucleosome	35 healthy controls 105 sepsis patients 60 asymptomatic HIV patients 126 patients with malaria	NR 64 U/mL NR 175 U/mL
de Jong et al. [79]	2014	Prospective Monocenter	Plasma	ELISA	H3 H2A, H 2B and dsDNA	82 healthy controls 44 sepsis [12 non-survivors]	Survivors 33.6 ± 4 U/mL * Non-survivors 192.3 ± 5 U/mL *
Zeerleder et al. [80]	2012	Retrospective Monocenter	Plasma	ELISA	H3 H2A, H 2B and dsDNA	38 children with meningococcal sepsis	47–8638 U/mL
Chen et al. [81]	2012	Prospective Multicenter [2 hospitals in China]	Plasma	ELISA [Roche]	Mono anti-histone Mono anti-DNA-POD	Medical: 45 sepsis vs. 29 controls [no sepsis] Post-surgery: 70 sepsis vs. 21 controls [no sepsis]	2.98 [0.30–12.60] vs. 1.29 [0.11–9.86] AU 1.86 [0.40–10.27] vs. 0.78 [0.35–9.69] AU
Weber et al. [82]	2008	Prospective Monocenter	Serum	ELISA [Roche]	Mono anti-histone Mono anti-DNA-POD	11 healthy volunteers 16 severe sepsis patients 10 ICU patients without sepsis	0.118 ± 0.036 AU 0.356 ± 0.057 AU 0.149 ± 0.026 AU
Zeerleder et al. [83]	2003	Retrospective Monocenter	Plasma	ELISA	H3 H2A, H 2B and dsDNA,x	14 fever 15 SIRS 32 severe sepsis 8 septic shock	38 [<35–285] units/mL 53 [<35–793] units/mL 269 [<35–1947] units/mL 814 [52–1979] units/mL

* Values on day 7; NR: not reported.

**Table 2 biomedicines-12-01385-t002:** Correlation of nucleosome levels with markers of neutrophil extracellular traps [NETs].

Author [Ref.]	NET Biomarker Utilized	Reported Correlation
van der Meer et al. [18]	elastase-a1-antitrypsin	r = 0.155 (*p* = 0.1295)
Morimont et al. [69]	citrullinated H3R8-nucleosomes, free citrullinated histones, NE and MPO	NE: Pearson r = 0.719
Duplessis et al. [73]	cfDNA	r = 0.41
Kaufman et al. [74]	human neutrophil elastase DNA	r^2^ = 0.3962 (*p* = 0.0499)
Delabranche et al. [75]	DNA-bound MPO	r^2^ = 0.397 (*p* = 0.004)
Huson et al. [78]	elastase-α1antitrypsin	r = 0.41 (*p* < 0.0001)
de Jong et al. [79]	neutrophil elastase	r = 0.84 (*p* < 0.001)
Zeerleder et al. [80]	elastase–α1antitrypsin complexes	r = 0.206 (*p* = 0.200)

NE: neutrophil elastase; MPO: myeloperoxidase.

**Table 3 biomedicines-12-01385-t003:** Nucleosome levels and sepsis severity.

Author [Ref.]	Sample Collection Time	Nucleosome Levels Predict Sepsis Severity	Severity Score		
			SOFA	APACHE II	SAPS II
Haem Rahimi et al. [66]	Daily (D1–D8)	Yes	r = 0.4 (*p* < 0.0001)	NR	r = 0.2 (*p* = 0.008)
Rai et al. [68]	Single (within 24 h of diagnosis)	No	0.08 (*p* = 0.46)	0.10 (*p* = 0.34)	NR
Morimont et al. [69]	Single (admission)	Yes	0.61	0.47	NR
Duplessis et al. [73]	Daily (T0 and T24)	Yes (a small correlation)	NR	0.24	NR
Kaufman et al. [74]	Within 24 h of admission and on day 4	No	R^2^ = 0.195 (*p* = 0.0362)	NR	NR
Miki et al. [77]	Days 0, 1, 3, 7	No	NR	NR	NR
Zeerleder et al. [80]	Days 0–8	Yes	R = 0.44 (*p* = 0.008)	NR	NR
Chen et al. [81]	Admission, days 1, 3, 5, 7	Yes	Admission r = 0.21 (*p* = 0.03)	Admission r = 0.24 (*p* = 0.01)	NR
Zeerleder et al. [83]	Admission	Yes	NR	NR	NR

SOFA: sequential organ failure assessment, and NR: not reported.

**Table 4 biomedicines-12-01385-t004:** Nucleosome levels and sepsis prognosis.

Author [Ref.]	Survivors	Non-Survivors	*p* Value [Survivors vs. Non-Survivors]	Prediction of Mortality
Haem Rahimi et al. [66]	1333.14 [385.14–3637.92] ng/mL	1919 [880.75–12,098.9] ng/mL	0.006	Cut-off 4639 ng/mL AUC 0.63
Eichhorn et al. [67]	0.09 [0.05; 0.11] AU	0.11 [0.05; 0.2] AU	NS	NR
Rai et al. [68]	185.0 [68.0–1721.0] pg/µL	345.0 [65.0–1584.2] pg/µL	0.004	Cut-off 215.0 [pg/μL] AUC: 0.68 [95% CI 0.56–0.80] Odds ratio 3.42 [1.35–8.68]
Morimont et al. [69]	785.2 [173.4–3076.1] ng/mL	901.6 [402.7–16,032.5] ng/mL	0.0664	NR
Duplessis et al. [73]	3.2 ± 9.1 µg/mL	5.0 ± 4.9 µg/mL	0.007	T0: AUC: 0.75 T24: AUC: 0.67
Kaufman et al. [74]	NR	NR	NR	Yes, Wald = 5.31
Raffray et al. [76]	9.2 AU	71.8 AU	*p* < 0.0001	NR [available from figure]
Miki et al. [77]	NR	NR	NS	Day 7 AUC: 0.57
Huson et al. [78]	60 [25–135] AU/mL	333 [298–456] AU/mL	0.0002	NR
de Jong et al. [79]	33.6 ± 4 AU/mL	192.3 ± 5 AU/mL	0.001	NR
Zeerleder et al. [80]	583 (47–2329) AU/mL	2244 [610–8638] AU/mL	0.0061	NR
Chen et al. [81]	1.97 AU	2.58 AU	0.06	NR
Zeerleder et al. [83]	276 [35–1947] AU/mL	628 [35–1979] AU/mL	0.333	NR

AUC: area under the receiver operating characteristic curve; AU: artificial unit; NR: not reported; and NS: not significant.

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
