# Peer review of "Circulating Nucleosomes as a Novel Biomarker for Sepsis: A Scoping Review"

_biomedicines, 2024, doi:10.3390/biomedicines12071385_

Round 1

Reviewer 1 Report

Comments and Suggestions for Authors

The authors address a very interesting topic. The early diagnosis of septic shock is a current and central topic in clinical and basic research

The study and expression of nucleosomes is very interesting, but difficult to reproduce in clinical practice. However, it is a very promising place to start from. Basic research has offered an incipit to clinical research for many years. The paper is well written, the statistical work is adequate and the conclusions leave open the possibility of further studies

However, the authors should include a paragraph on biomarkers used in clinic. The paper is a review and therefore must address all aspects of the early diagnosis of sepsis.

Author Response

We thank the reviewer for their comments and one paragraph described biomarkers used in clinic is added in the introduction part.

Biomarkers provide critical insights into the systemic manifestations of sepsis, such as host-response indicators like C-reactive protein (CRP) and procalcitonin (PCT), organ dysfunction (e.g., kidney injury biomarkers), and microbiological documentationPedro Póvoa et al., Intensive Care Med (2023) 49:142–153. Additionally, they facilitate the tailoring of therapies to specific patients and monitoring therapeutic effectsRello J, van Engelen TSR, Alp E, Calandra T, Cattoir V, Kern WV, et al., Clin Microbiol Infect. 2018;24(12):1264–72. Despite the identification of over 250 new sepsis biomarkers, the complexity of sepsis and the enhanced sensitivity of various detection techniques indicate that specific diagnostic biomarkers and effective therapeutic approaches for sepsis remain elusive in clinical practice and require extensive validation.

Reviewer 2 Report

Comments and Suggestions for Authors

Authors performed scoping review about circulating Nucleosomes as a Novel Biomarker for Sepsis. Nucleosomes are not well-known area in research, and in that sense, this scoping review might be helpful.

I have a few comments.

Line 51-54 was commented in line 37-40, so it should be deleted or otherwise organized.

In line 148, early survival after transplantation should be changed to early mortality after transplantation.

In table 2 and 3, p value should be reported in every study.

In table 4, The columns of sampling time and number of patients might be moved before survivors and non-survivors because p value should follow immediately after columns of survivors and non-survivors.

Author Response

We thank the reviewer for their comments and have made revisions accordingly.

Line 51-54 was commented in line 37-40, so it should be deleted or otherwise organized.

The reviewer is right, actually line 51-54 is belong to fingure 1 lengend.

In line 148, early survival after transplantation should be changed to early mortality after transplantation.

Change has been made.

In table 2 and 3, p value should be reported in every study.

Reported p value has been added inside table.

In table 4, The columns of sampling time and number of patients might be moved before survivors and non-survivors because p value should follow immediately after columns of survivors and non-survivors.

The columns of sampling time and number of patients have been removed.

Reviewer 3 Report

Comments and Suggestions for Authors

In this study, author identified circulating nucleosomes as a novel biomarker for sepsis. They discussed the possible role of nucleosome concentrations in the diagnosis, prognosis, and therapeutic management of sepsis. A comprehensive literature search of the Cochrane and Medline libraries from 1996 to February 1, 2024, identifed 110 potentially eligible studies, of which 19 met the inclusion criteria, encompassing a total of 39 SIRS, 893 sepsis patients, 280 septic shock, 117 other ICU control patients, and 345 healthy volunteers. Enzyme-linked immunosorbent assay [ELISA] was the primary method of nucleosome measurement. Studies consistently reported significant correlations between nucleosome levels and other NET biomarkers. Nucleosome levels were higher in patients with sepsis than in healthy volunteers, and associated with disease severity, as indicated by SOFA and APACHE II scores. Non-survivors had higher nucleosome levels than survivors. Circulating nucleosome levels therefore show promise as early markers of NETosis in sepsis, with moderate diagnostic accuracy and strong correlations with disease severity and prognosis. However, the available evidence is drawn mainly from single center, observational studies with small sample sizes and varied detection methods, warranting further investigation. In general, this review is interesting and well-written. Here are some comments from this reviewer:

1. How about the roles of nucleosome in macrophages?

2. Is nucleosome correlated with negative feedback control factor such as SOCS1?

3. How about the functions of nucleosome in sepsis-related animal model such CLP?

Author Response

We thank the reviewer for their comments and have made revisions accordingly.

  1. How about the roles of nucleosome in macrophages?

Macrophages, as innate immune cells, release nucleosomes upon activation and subsequent cell death, such as during pyroptosis (reference 22) as noted in lines 86-90. Furthermore, macrophage cell death in sepsis has been documented (references 27, 28) in lines 98-100.

  1. Is nucleosome correlated with negative feedback control factor such as SOCS1?

The reviewer raised an important question that warrants further investigation. To my knowledge, there is no specific study designed to answer this question. However, currently, there are three categories of drug targets for histones in the pharmaceutical field (Silk E, et al. Cell Death and Disease (2017) 8, e2812): 1) compounds that directly neutralize histones, 2) agents that block histone release, such as DNase1, and 3) signaling inhibitors, such as TLR-blocking monoclonal antibodies.

Suppressor of cytokine signaling 1 (SOCS1) is a critical regulator of cytokine signaling and immune responses. It functions as a negative regulator of toll-like receptor (TLR)-induced inflammatory signaling and as a regulator of metabolic reprogramming, preventing overwhelming inflammatory responses and organ damage during sepsis (Annie Rocio Piñeros Alvarez, et al., JCI Insight. 2017 Jul 6;2(13)). Based on this information, nucleosome levels should be correlated with negative feedback control factors such as SOCS1.

  1. How about the functions of nucleosome in sepsis-related animal model such CLP?

In lines 84-85, we noted that nucleosome levels increase in both the CLP sepsis model and in healthy volunteers challenged with LPS, indicating an inflammatory response due to innate immune cell activation. Furthermore, in lines 159-162, we reported that anti-histone therapy could reduce the elevated nucleosome levels in a clinically relevant sepsis model. These findings suggest that nucleosome levels could serve as a biomarker for NETosis in the CLP sepsis model, with concentrations correlating to sepsis severity and prognosis. Additionally, nucleosome levels could guide sepsis therapy targets, warranting further investigation.

Round 2

Reviewer 1 Report

Comments and Suggestions for Authors

The authors made the requested changes